# Poly(ADP-ribose) polymerase 1 searches DNA via a 'monkey bar' mechanism

Johannes Rudolph[1], Jyothi Mahadevan[1], Pamela Dyer[1,2], Karolin Luger[1,2]*

[1]Department of Chemistry and Biochemistry, University of Colorado Boulder, Boulder, United States; [2]Howard Hughes Medical Institute, University of Colorado Boulder, Boulder, United States

**Abstract** Poly(ADP-ribose) polymerase 1 (PARP1) is both a first responder to DNA damage and a chromatin architectural protein. How PARP1 rapidly finds DNA damage sites in the context of a nucleus filled with undamaged DNA, to which it also binds, is an unresolved question. Here, we show that PARP1 association with DNA is diffusion-limited, and release of PARP1 from DNA is promoted by binding of an additional DNA molecule that facilitates a 'monkey bar' mechanism, also known as intersegment transfer. The WGR-domain of PARP1 is essential to this mechanism, and a point mutation (W589A) recapitulates the altered kinetics of the domain deletion. Demonstrating the physiological importance of the monkey bar mechanism for PARP1 function, the W589A mutant accumulates at sites of DNA damage more slowly following laser micro-irradiation than wild-type PARP1. Clinically relevant inhibitors of PARP1 did not alter the rate or mechanism of the release of PARP1 from DNA.

DOI: https://doi.org/10.7554/eLife.37818.001

## Introduction

Poly(ADP-ribose) polymerase 1 (PARP1) serves as a first responder to DNA damage and is the founding member and most abundant representative of the large family of diphtheria toxin-like ADP-ribosyltransferases (ARTDs) (*Bai, 2015*; *Bock and Chang, 2016*; *Beck et al., 2014*; *Daniels et al., 2015*; *De Vos et al., 2012*; *Mashimo et al., 2014*; *Morales et al., 2014*). Binding to either single or double-strand DNA breaks (SSBs or DSBs) enzymatically activates PARP1 to use NAD$^+$ in polymerizing long chains of poly(ADP)-ribose (PAR) onto itself and other nuclear acceptor proteins such as histones and DNA repair proteins. These PAR chains then recruit the appropriate DNA repair machinery containing PAR-binding motifs (*Karlberg et al., 2013*; *Teloni and Altmeyer, 2016*). PARP1 is of special interest because it is a validated target for cancer therapy (*Tangutoori et al., 2015*; *Liu et al., 2014*). Most notably, olaparib and rucaparib are in clinical use for treatment of ovarian and/or breast cancer in BRCA1/2 negative patients, and there are many on-going phase III clinical trials for inhibitors of PARPs either as monotherapy or in combination with chemo- or radiotherapy.

Overall, the domain structures of the 16 members of the ARTD family are quite diverse, but they all share a common catalytic core domain (~40 kDa) (*Barkauskaite et al., 2015*). Clinically relevant inhibitors of PARP1 bind in the catalytic domain. The N-terminal region of PARP1 contains five additional domains; three Zn-finger domains, an automodification domain that contains a BRCT-fold, and a WGR domain (*Figure 1A*). Seminal work from the Pascal laboratory has provided a molecular understanding of how Zn1, Zn2, Zn3, and the WGR domain collaborate to recognize DNA strand breaks in a structure-specific and sequence-independent manner, and subsequently activate the catalytic activity of PARP1. Zn1 and Zn2 separately (*Langelier et al., 2011a*), and together in the context of an SSB (*Eustermann et al., 2015*), bind one DNA end each using two points of contact, termed the phosphate backbone grip and the base stacking loop. In the context of a DSB, this grip-loop interaction mode is maintained by Zn1, while the Zn3 and WGR domains make additional

*For correspondence:
karolin.luger@colorado.edu

**Competing interests:** The authors declare that no competing interests exist.

**eLife digest** Our cells constantly withstand damage that can lead to breaks in the strands of our DNA. These cuts need to be fixed for the cell to stay healthy. When a break happens, one of the first responders to the scene is a protein known as PARP1. It binds to the ruptured strand (or strands) and then it recruits other repair agents to that location. But first, PARP1 needs to scan for cuts and notches amongst an overwhelming amount of DNA that is still intact. This is a complicated task, especially since the protein tends to bind both broken and unbroken DNA. How does it not stay 'stuck' on an undamaged portion of the genome?

Here, Rudolph et al. use a combination of biochemical techniques and cell biology to show that PARP1 travels through our genome by swinging from one DNA location to another, the way a child grabs onto monkey bars. One of the DNA-binding domains of PARP1, known as the WGR-domain, acts like an arm and initiates the movement by gripping onto a new segment of DNA. In fact, chopping off the WGR-domain or disabling it through mutations makes PARP1 worse at finding DNA breakages in the cell.

Unfixed DNA damage can lead to a cell becoming cancerous; ultimately, if the breakages keep accumulating, the cell does not survive. This makes PARP1 an important target for cancer treatment. Indeed, certain drugs already rely on trapping the protein so that tumor cells die. Understanding how cells cope with DNA damage and exactly how PARP1 works could help in the fight against cancer.

DOI: https://doi.org/10.7554/eLife.37818.002

contacts to the DNA (*Figure 1B*) (*Langelier et al., 2012*). Importantly, stepwise assembly of the different domains of PARP1 on DNA leads to the destabilization of the helical subdomain (HD) of the catalytic domain, which results in activation of its ADP-ribosyl transferase activity (*Eustermann et al., 2015*; *Langelier et al., 2018*; *Dawicki-McKenna et al., 2015*).

In cells, PARP1 is known to contribute to many types of DNA repair mechanisms, including base excision repair, homologous recombination, nucleotide excision repair, and alternative non-homologous end-joining (*de Murcia et al., 1997*). In vitro, PARP1 is activated by a wide variety of DNA damage models including nicks, gaps, blunt ends, 5'- or 3'- extensions, all with or without a 5'-phosphate (*Langelier et al., 2014*). There is now clear evidence from multiple laboratories that PARP1 also binds tightly to undamaged DNA. For example, the Kraus laboratory has shown that PARP1 binds to and condenses intact chromatin, represses Pol II-dependent transcription, and is activated for auto-PARylation (*Kim et al., 2004*). We have previously shown that PARP1 serves as a chromatin architectural protein and interacts tightly ($K_d$ ~nM) with and is activated by various nucleosome constructs (*Clark et al., 2012*; *Muthurajan et al., 2014*). Additionally, atomic force microscopy has shown that PARP1 binds not only to DNA ends or specific nicks, but also has significant affinity for undamaged DNA (*Sukhanova et al., 2016*). Most recently, single molecule tightrope assays have demonstrated that PARP1 interacts with and moves along undamaged DNA (*Liu et al., 2017*).

Thus PARP1 faces a similar 'speed-stability' paradox (*Mirny et al., 2009*; *Zandarashvili et al., 2015*; *Halford and Marko, 2004*) as transcription factors that need to find their target recognition site in an overwhelming excess of non-specific sites for which they also have significant affinity. PARP1 must rapidly search the genome for damaged DNA, yet it has significant affinity for the billions of base pairs of undamaged DNA that are present at concentrations of ~100 mg/mL in the nucleus (*Krebs et al., 2017*). In fact, laser micro-irradiation experiments in live cells have shown that PARP1 significantly accumulates at DNA damage sites in less than 10 s (*Mortusewicz et al., 2007*). The conundrum is that repeated cycles of release of PARP1 from undamaged DNA, random diffusional collisions, and rebinding to a different location may not be fast enough to explain how PARP1 can rapidly localize to sites of DNA damage. Various models have been put forth and tested for explaining how 'facilitated diffusion' could accelerate this search process, all of which recognize the importance, as opposed to hindrance, of non-specific binding to DNA for efficient site localization (*Halford and Marko, 2004*; *Berg et al., 1981*; *Iwahara et al., 2006*; *Doucleff and Clore, 2008*). These models include binding followed by one-dimensional sliding along DNA, hopping to a nearby site in the same chain, and intersegment transfer via an intermediate loop that is formed when

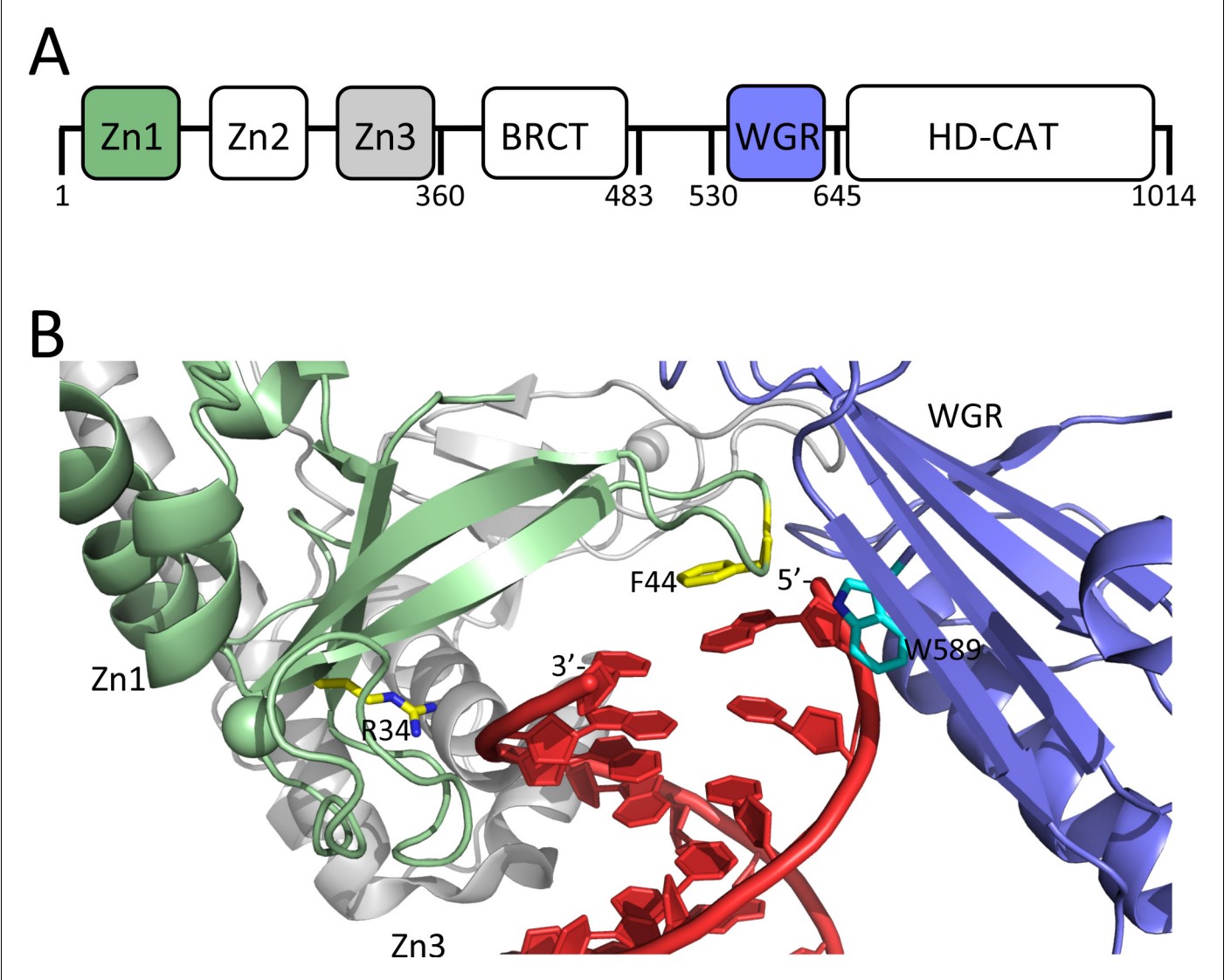

**Figure 1.** Domain organization of PARP1 and structural details of how PARP1 binds to a DSB. (**A**) Schematic of the domains of PARP1; (**B**) DNA-binding domains (Zn1, green, Zn3 gray, and WGR, blue) of PARP1 engaging a DNA DSB (red). Residues R34 and F44 of the phosphate backbone grip and the base stacking loop in the Zn1 domain are shown in yellow and W589 in the WGR domain is shown in light blue. Coordinates were taken from 1dqy.
DOI: https://doi.org/10.7554/eLife.37818.003

The following figure supplement is available for figure 1:

**Figure supplement 1.** SDS-PAGE showing purified PARP1, its deletion constructs and the W589A point mutant.
DOI: https://doi.org/10.7554/eLife.37818.004

the protein binds two different DNA sites at the same time. While some localized sliding along DNA has been reported for PARP1 (*Liu et al., 2017*), a more thorough kinetic characterization of binding to and dissociation from DNA is needed in order to address how PARP1 can efficiently localize to sites of DNA damage to initiate repair.

PARP1 has been found to associate more tightly with DNA in vivo in the presence of clinically relevant inhibitors. This phenomenon, known as PARP 'trapping' (*Tangutoori et al., 2015*; *Brown et al., 2016*; *Shen et al., 2015*; *Murai et al., 2012*; *Pommier et al., 2016*) is thought to be in part responsible for the clinical effects of PARP inhibitors and has been used to explain the numerous discrepancies between in vitro inhibition of PARP1 vs. potency in preclinical models. For example, talazoparib is 100-fold more potent at trapping PARP1 on DNA and >50 fold more potent at killing cancer cells than rucaparib and olaparib, although the apparent IC-50's for all three compounds are quite similar (1–5

**Scheme 1.** Kinetic model for association of PARP1 with DNA.
DOI: https://doi.org/10.7554/eLife.37818.018

nM) (*Shen et al., 2015*). Further complicating matters, extensive biochemical investigations of PARP1 trapping failed to provide evidence for an allosteric interaction between DNA- and inhibitor-binding (*Hopkins et al., 2015*), suggesting that trapping is due solely to inhibition of catalytic activity (but see [*Langelier et al., 2018*]). Thus, an evaluation of PARP inhibitors in a quantitative assay that measures DNA binding and release has the potential to shed further light on this controversial issue.

Here, we report on the kinetics of association and dissociation of PARP1 with DNA. We find that association of PARP1 with DNA is extremely fast, and that dissociation depends on the formation of a ternary complex where a second DNA molecule binds before release of the original DNA. We find that the WGR-domain, more specifically the conserved residue Trp589, is essential for triggering DNA-dependent release of DNA from PARP1, and we demonstrate the importance of this mechanism of DNA release for the accumulation of PARP1 at sites of DNA damage in the cell. Finally, we find that clinically relevant inhibitors do not perturb the rate or mechanism of release of DNA from PARP1.

## Results

### Association of PARP1 with DNA is extremely fast

We began our investigations by measuring the rate of association of PARP1 with DNA. Varying concentrations of PARP1 (60–250 nM) were mixed in a stopped-flow apparatus with fixed concentrations (30 nM) of a fluorescently labeled model of a double-strand break with a 5'-phosphate (p18mer*). Addition of protein results in an increase in fluorescence anisotropy that is not observed by addition of buffer alone (*Figure 2A*). The data at all concentrations of PARP1 could be fitted with a single exponential to yield $k_{obs}$ with very good residuals (*Figure 2A*). Under idealized experimental conditions wherein the concentration of PARP1 greatly exceeds the concentration of p18mer*, one would expect a replot of $k_{obs}$ vs. the concentration of PARP1 to yield a straight line, as was indeed observed here (*Figure 2A*, inset). The slope of such a line equals the apparent second order rate constant of association, whereas the y-intercept equals the first-order rate constant of dissociation. To analyze the data more rigorously, we used Kintek Explorer, a powerful fitting program that allows for global model-dependent fitting that does not require adherence to limiting conditions (*Figure 2B*). Our analysis yields a $k_1$ of 3.1 $nM^{-1}s^{-1}$ (*Scheme 1*, *Table 1*), which is significantly greater than previously reported for PARP1 associating with DNA as measured using surface plasmon resonance (*Jorgensen et al., 2009*). The rate of dissociation ($k_{-1}$) could not be determined from this experiment since no significant dissociation occurs over the 25 ms time course of the experimental observation. Using global fitting, we could derive an upper bound for $k_{-1}$ of 10 $s^{-1}$. Thus, the true equilibrium dissociation constant ($K_D$) of a double-strand break under these conditions is <3 nM (*Figure 4—figure supplement 2*), lower than the previously reported $K_D$s of 31 nM (*Clark et al., 2012*), 14 nM (*Langelier et al., 2010*), and 97 nM (*Langelier et al., 2018*) (see Discussion).

**Scheme 2.** Kinetic model for dissociation of PARP1 from labeled dDNA in the presence of competing unlabeled DNA.
DOI: https://doi.org/10.7554/eLife.37818.021

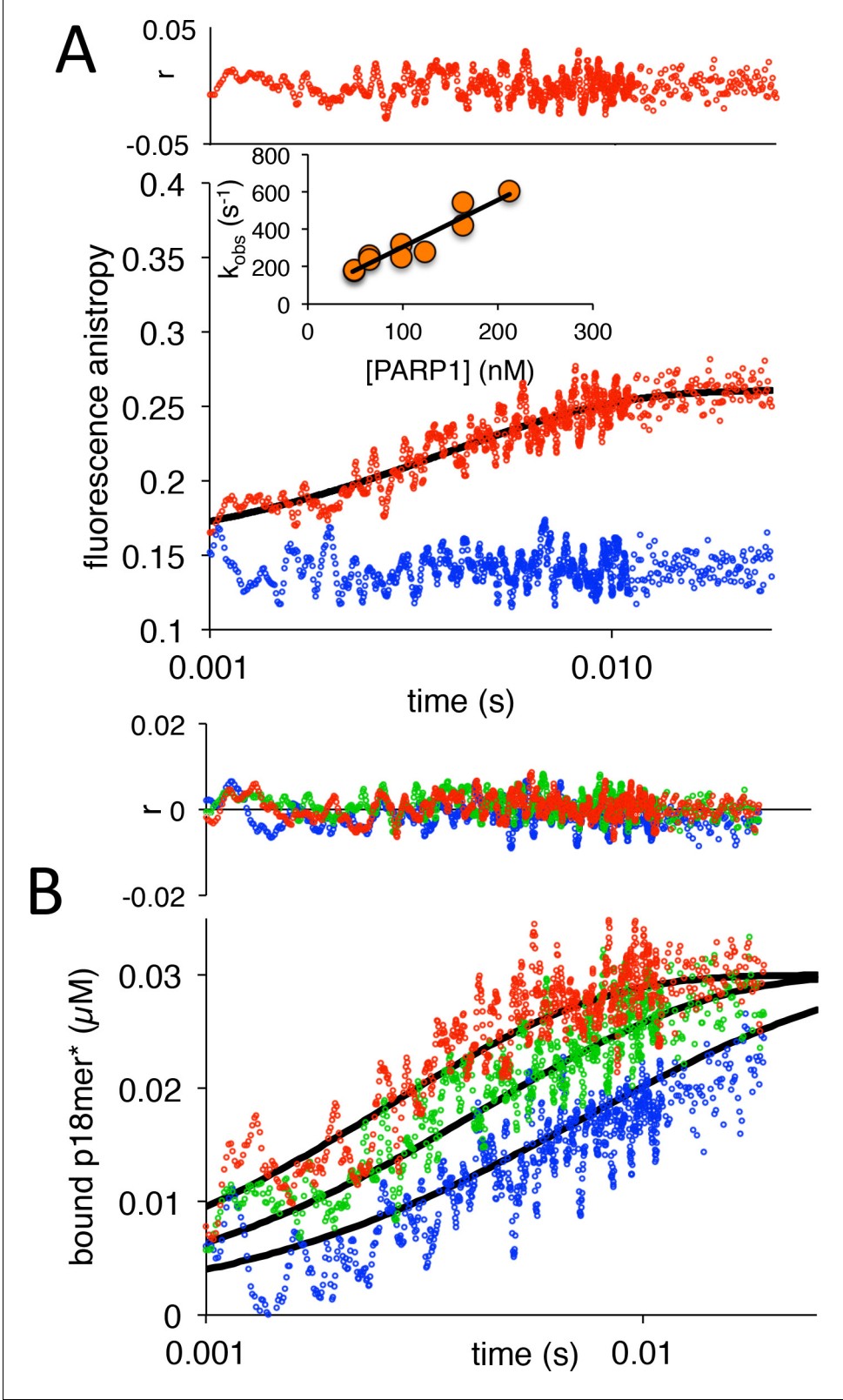

**Figure 2.** PARP1 association with DNA as monitored by fluorescence anistropy. (**A**) Representative measurement of PARP1 association with DNA as monitored by fluorescence anisotropy. Shown are the data in the absence of PARP1 (blue) and in the presence of 83 nM PARP1 (red). The black line shows a first-order exponential fit to the data and the residuals (r) from this fit are shown above. The inset shows a replot of $k_{obs}$ vs. varying concentrations

*Figure 2 continued on next page*

*Figure 2 continued*

of PARP1. (B) Global fitting of three representative concentrations of PARP1 using the mechanism in *Scheme 1*: 50 nM in blue, 83 nM in green and 133 nM in red. Residuals (r) for the three concentrations are shown in the corresponding colors above.

DOI: https://doi.org/10.7554/eLife.37818.005

The following source data and figure supplements are available for figure 2:

**Source data 1.** Raw data for representative measurement of PARP1 association with DNA as monitored by fluorescence anisotropy as shown in panel A.
DOI: https://doi.org/10.7554/eLife.37818.016
**Source data 2.** Global fitting of three representative concentrations of PARP1 using the mechanism in *Scheme 1* with raw data, best fits, and residuals as shown in panel B.
DOI: https://doi.org/10.7554/eLife.37818.017
**Figure supplement 1.** Association of ΔZn1 with p18mer*.
DOI: https://doi.org/10.7554/eLife.37818.006
**Figure supplement 1—source data 1.** Association of ΔZn1 with p18mer*.
DOI: https://doi.org/10.7554/eLife.37818.007
**Figure supplement 2.** Association of ΔZn2 with p18mer*.
DOI: https://doi.org/10.7554/eLife.37818.008
**Figure supplement 2—source data 1.** Association of ΔZn2 with p18mer*.
DOI: https://doi.org/10.7554/eLife.37818.009
**Figure supplement 3.** Association of ΔZn3 with p18mer*.
DOI: https://doi.org/10.7554/eLife.37818.010
**Figure supplement 3—source data 1.** Association of ΔZn3 with p18mer*.
DOI: https://doi.org/10.7554/eLife.37818.011
**Figure supplement 4.** Association of ΔWGR with p18mer*.
DOI: https://doi.org/10.7554/eLife.37818.012
**Figure supplement 4—source data 1.** Association of ΔWGR with p18mer*.
DOI: https://doi.org/10.7554/eLife.37818.013
**Figure supplement 5.** Association of W589A with p18mer*.
DOI: https://doi.org/10.7554/eLife.37818.014
**Figure supplement 5—source data 1.** Association of W589A with p18mer*.
DOI: https://doi.org/10.7554/eLife.37818.015

## Dissociation of PARP1 from DNA requires binding of a second DNA molecule

Because we were unable to determine the rate of DNA dissociation from PARP1 in the previous experiment, we designed an experiment to explicitly measure this rate using competition. Here, we pre-form a complex between PARP1 and fluorescently labeled DNA and use an excess of unlabeled DNA to compete away the labeled DNA and prevent its re-association with PARP1. We began these investigations by first performing a label-swap experiment to ensure that unlabeled p18mer behaves similarly to fluorescein labeled p18mer*. Since the experimental read-out is based on the change in fluorescence anisotropy of p18mer*, we used a fixed and limiting concentration of total labeled DNA such that no excess p18mer* is present. PARP1 (37 nM), pre-bound to either p18mer or p18mer* (25 nM) was mixed with 25 nM p18mer* or p18mer (respectively) in a stopped-flow

$$\text{P1•DNA*} \underset{k_{-2}}{\overset{k_2[\text{DNA}]}{\rightleftharpoons}} \text{DNA• P1•DNA*} \underset{[\text{DNA*}] k_{-3}}{\overset{k_3}{\rightleftharpoons}} \text{P1•DNA}$$

**Scheme 3.** Kinetic model for the dissociation of PARP1 from labeled DNA that depends on formation of a ternary complex with the unlabeled DNA.

DOI: https://doi.org/10.7554/eLife.37818.026

**Table 1.** Kinetic parameters for DNA association and dissociation

All values (mean and standard deviation) were derived from global fitting in Kintek Explorer of data from at least three independent experiments performed on different days and using at least two independent preparations of protein. All $k_1$ rate constants were derived using the kinetic model in *Scheme 1*. All other rate constants came from using the kinetic model in *Scheme 3*, except for the $k_{-1}$ values for ΔWGR and W589A, which were derived using *Scheme 2*. *The value of $k_2$ assumes the full range of 5–450 possible binding sites per 4.5 kb plasmid (see Results).

| | $k_1$ (nM$^{-1}$s$^{-1}$) | $k_{-1}$ (s$^{-1}$) | $k_2$ (nM$^{-1}$s$^{-1}$) | $k_{-2}$ (s$^{-1}$) | $k_3$ (s$^{-1}$) | $k_{-3}$ (nM$^{-1}$s$^{-1}$) |
|---|---|---|---|---|---|---|
| WT | 3.1 ± 0.2 | <10 | 0.043 ± 0.019 | 102 ± 22 | 9.7 ± 0.8 | 0.013 ± 0.002 |
| ΔZn1 | 3.7 ± 0.4 | <10 | 0.018 ± 0.012 | 97 ± 19 | 21 ± 5 | 0.027 ± 0.009 |
| ΔZn2 | 4.0 ± 1.3 | <10 | 0.073 ± 0.015 | 145 ± 66 | 15 ± 4 | 0.023 ± 0.007 |
| ΔZn3 | 5.5 ± 0.3 | <10 | 0.034 ± 0.013 | 76 ± 66 | 19 ± 7 | 0.050 ± 0.016 |
| ΔWGR | 2.4 ± 1.1 | 18.7 ± 2.7 | | | | |
| W589A | 4.2 ± 0.5 | 20.2 ± 2.4 | | | | |
| WT (with intact plasmid) | | | 0.041–3.7 | 161 ± 8 | 9 ± 3 | 0.032 ± 0.006 |

DOI: https://doi.org/10.7554/eLife.37818.019

The following source data is available for Table 1:

Source data 1. Kinetic parameters for DNA association and dissociation

DOI: https://doi.org/10.7554/eLife.37818.020

apparatus. Dissociation of p18mer or p18mer* (followed by binding of the competitor) was monitored by an increase or decrease in fluorescence anisotropy, respectively (*Figure 3—figure supplement 1*). The similarity of these two experiments is best visualized by plotting the sum of the signal to generate a flat line equal to the probe concentration (25 nM), a pseudo-residual indicating that p18mer and p18mer* are kinetically indistinguishable in our assay.

In order to probe the mechanism of DNA dissociation from PARP1, we next varied the concentrations of competitor DNA. Under ideal experimental conditions, where the concentration of competitor DNA (>500 nM) greatly exceeds the probe concentration (25 nM), and assuming the simplest model wherein the rate of dissociation is rate-determining ($k'_1$ [DNA]>>$k_{-1}$, *Scheme 2*), we expect that $k_{obs}$ would be independent of the concentration of competitor DNA. PARP1 (37 nM), pre-bound to p18mer* (25 nM), was mixed with various concentrations of competitor DNA (p18mer, 500 nM – 4000 nM) in a stopped-flow apparatus and dissociation of p18mer* was monitored by a decrease in fluorescence anisotropy (*Figure 3*). The data could be fitted to a single exponential to yield $k_{obs}$ with very good residuals (*Figure 3*). However, as seen in the data in *Figure 3* by comparing dissociation in the presence of 2.2 vs 4 μM DNA, and in the replot of $k_{obs}$ vs. multiple concentrations of competitor DNA, $k_{obs}$ increases at increasing concentrations of competitor DNA (*Figure 3*, inset). Additionally, attempts to fit these data with *Scheme 2* in Kintek Explorer yielded very poor fits and highly skewed residuals (*Figure 4—figure supplement 2*). Thus, a different kinetic scheme is needed to fit these data, one where competitor DNA is actively contributing to the dissociation of the pre-bound p18mer*.

The simplest model to explain active participation of a competitor DNA in the dissociation of an already bound DNA is formation of a ternary complex wherein the competing DNA binds to PARP1 prior to the dissociation of the pre-bound DNA (*Scheme 3*). This model consists of four rate constants: $k_2$, (formation of the ternary complex), $k_{-2}$ (release of the competing DNA to regenerate the pre-bound complex), $k_3$ (release of the pre-bound DNA to generate PARP1 only bound to the competing DNA), and $k_{-3}$ (re-formation of the ternary complex). Experimentally, both the starting pre-bound complex and the ternary complex are assigned a high anisotropy, whereas the final complex bound only to competing, unlabeled DNA is assigned a low anisotropy. In order to best constrain the four rate constants required to describe *Scheme 3*, we used a broader range of competing DNA concentrations (50 nM – 4000 nM). Also, each concentration series was independently determined and fitted using Kintek Explorer at least three times. Representative fits of this model to the data are shown in *Figure 4*, and the residuals indicate very good agreement between the data and this model, even at low concentrations of competitor DNA where $k_{obs}$ does not fit to a simple

exponential and the apparent extent of exchange is significantly lower than at high concentrations. The aggregated rate constants are shown in *Table 1* and the derived dissociation constants are shown in *Figure 4—figure supplement 1*. The quality of the fits with the kinetic model in *Scheme 3* provides strong support for the requisite formation of a ternary complex in the dissociation of DNA from PARP1.

The second order rate constant of association for the second DNA molecule is 0.043 nM$^{-1}$s$^{-1}$ is almost two orders of magnitude lower than that for association of the first DNA oligomer. The $K_D$ for the second DNA strand is 2600 nM, explaining why this complex would be rarely if ever detected under typical experimental conditions performed at nanomolar concentrations of PARP1. Note that the rates of association and dissociation for the second DNA are not 'symmetrical' (i.e. $k_2 \neq k_{-3}$ and $k_{-2} \neq k_3$). This asymmetry is most pronounced in the comparison between $k_{-2}$ and $k_3$: the pre-bound DNA is less likely to dissociate than the second competitor DNA. This observation makes intuitive sense in that the newly incoming DNA presumably binds to a different (weaker) site than the originally more tightly bound DNA. Although there is a lack of symmetry in the rate constants, the kinetically derived dissociation constants ($K_D$s) are quite similar (*Figure 4—figure supplement 1*).

## The WGR domain provides the binding site for the second DNA molecule

Formation of a ternary complex with two different DNA molecules bound simultaneously requires two separate DNA binding sites. PARP1 has four domains that are known to contribute to DNA binding: Zn1, Zn2, Zn3, and WGR (*Figure 1*). In order to identify if one or more of these domains selectively contributes to the formation of the ternary complex required for efficient DNA release, we generated constructs of PARP1 lacking each of these individual domains. To facilitate proper assembly of the remaining domains, we inserted a flexible 30 amino acid linker into each deletion, except for the N-terminal deletion of Zn1. All mutants were purified to near homogeneity and were tested for DNA-dependent PARylation activity (*Figure 1—figure supplement 1*). As previously reported (*Langelier et al., 2012*), Zn1, Zn3, and WGR are essential for catalytic activity, and thus deletion of these domains disrupts PARylation activity. On the other hand, the deletion of the non-essential Zn2 domain does not affect PARylation activity.

We next measured the rates of association to, and dissociation from p18mer* for each of the individual deletions of the DNA-binding domains, using the stopped-flow anisotropy assays described above. As for wild-type PARP1, each deletion construct was assayed at multiple different concentrations of protein or DNA, and the data were analyzed using global fitting in Kintek Explorer. Despite each construct missing one DNA-binding domain, all four bound to DNA with similar second-order rate constants ($k_1$s, *Table 1*, *Figure 2—figure supplements 1–4*). In the dissociation experiment, the Zn deletions (ΔZn1, ΔZn2, and ΔZn3) behaved essentially like wild-type PARP1: increasing competitor DNA concentrations yielded increasing $k_{obs}$, and the data were best described by global fitting of the kinetic model of *Scheme 3* with the formation of a ternary complex (*Figure 4—figure supplements 3–5*, *Table 1*). In contrast, the ΔWGR mutant behaved dramatically differently; increasing concentrations of competitor DNA did not yield higher $k_{obs}$s, and globally the data were best described by *Scheme 2* (*Figure 5A*, *Table 1*). In the structure of PARP1 bound to a DSB, Trp589 in the WGR domain stacks against the ribose sugar of the 5'-end of the DNA (*Figure 1*) (*Langelier et al., 2012*). Since deletion of the entire WGR domain disrupted formation of the ternary complex, we tested whether the more conservative W589A substitution could recapitulate this effect. PARP1-W589A was prepared (*Figure 1—figure supplement 1*) and tested in both the association and dissociation assays. The W589A point mutation is properly folded, as the mutant and wild-type PARP1 have identical melting temperatures (43.9 ± 0.3 vs. 43.3 ± 0.3°C). Similar to what was observed with the deletion of the entire WGR domain, the W589A mutant also bound to free DNA rapidly (*Figure 2—figure supplement 5*), and released DNA via the simple mechanism in *Scheme 2* that is not dependent on binding a second DNA molecule (*Figure 5B*, *Table 1*).

## Undamaged DNA also facilitates the monkey-bar mechanism

If PARP1 movement around the nucleus is indeed facilitated by the monkey-bar mechanism, then undamaged DNA, not just a short oligomer, should also promote release of pre-bound DNA. To address this question, we used plasmid DNA as a competitor of p18mer* pre-bound to PARP1.

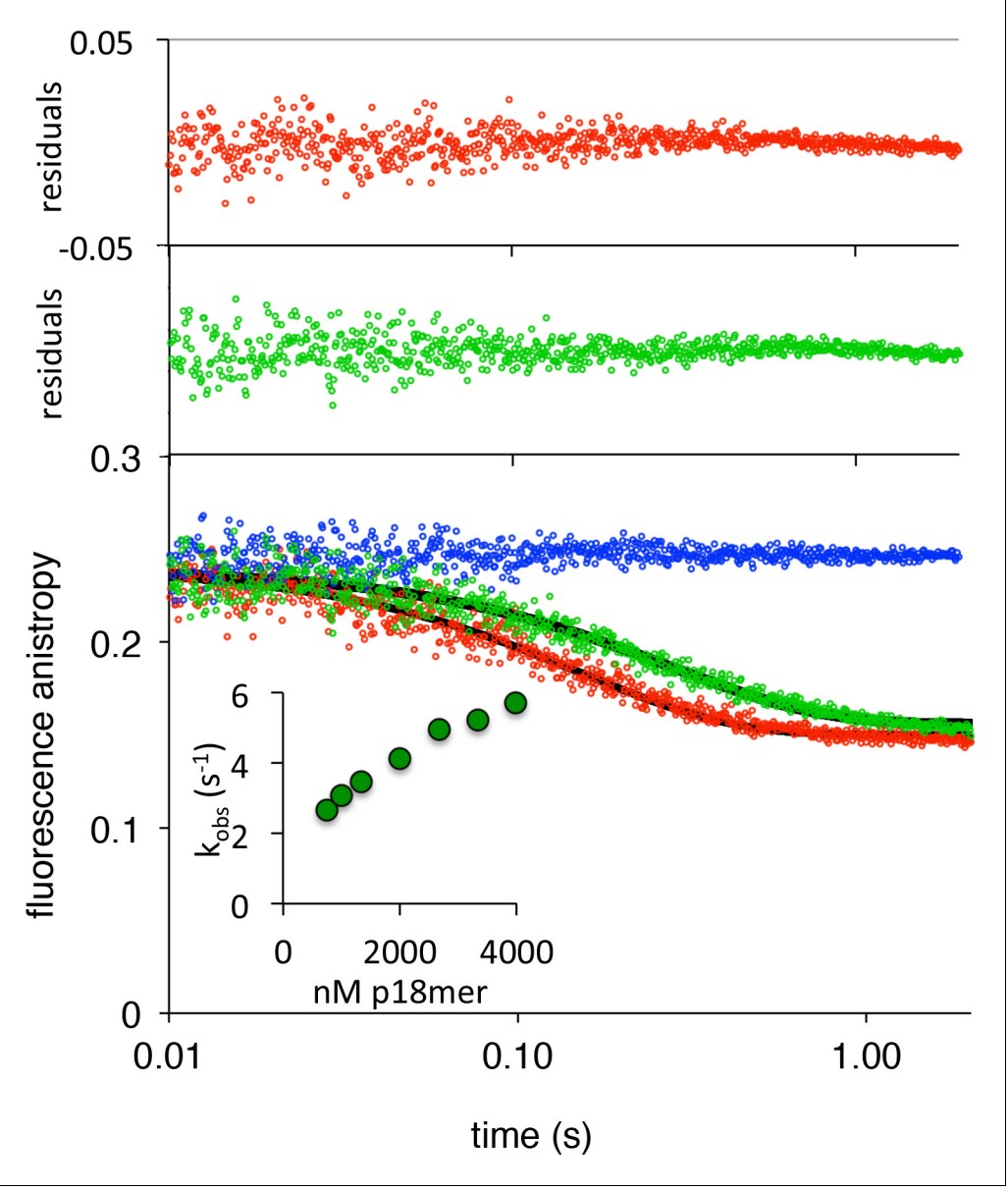

**Figure 3.** Representative measurement of PARP1 dissociation from DNA as monitored by fluorescence anisotropy. Shown are the data in the absence of competitor DNA (in blue) and in the presence of 2.2 μM (green) and 4 μM DNA (red). The black line shows a first-order exponential fit to the data and the residuals from these fits are shown in the corresponding colors above. The inset shows a replot of $k_{obs}$ vs. varying concentrations of competitor DNA.
DOI: https://doi.org/10.7554/eLife.37818.022

The following source data and figure supplements are available for figure 3:

**Source data 1.** Representative measurement of PARP1 dissociation from DNA as monitored by fluorescence anisotropy.
DOI: https://doi.org/10.7554/eLife.37818.025

**Figure supplement 1.** Label-swap experiment demonstrating that monitoring p18mer* and p18mer release are kinetically identical.
DOI: https://doi.org/10.7554/eLife.37818.023

**Figure supplement 1—source data 1.** Label-swap experiment demonstrating that monitoring p18mer* and p18mer release are kinetically identical.
DOI: https://doi.org/10.7554/eLife.37818.024

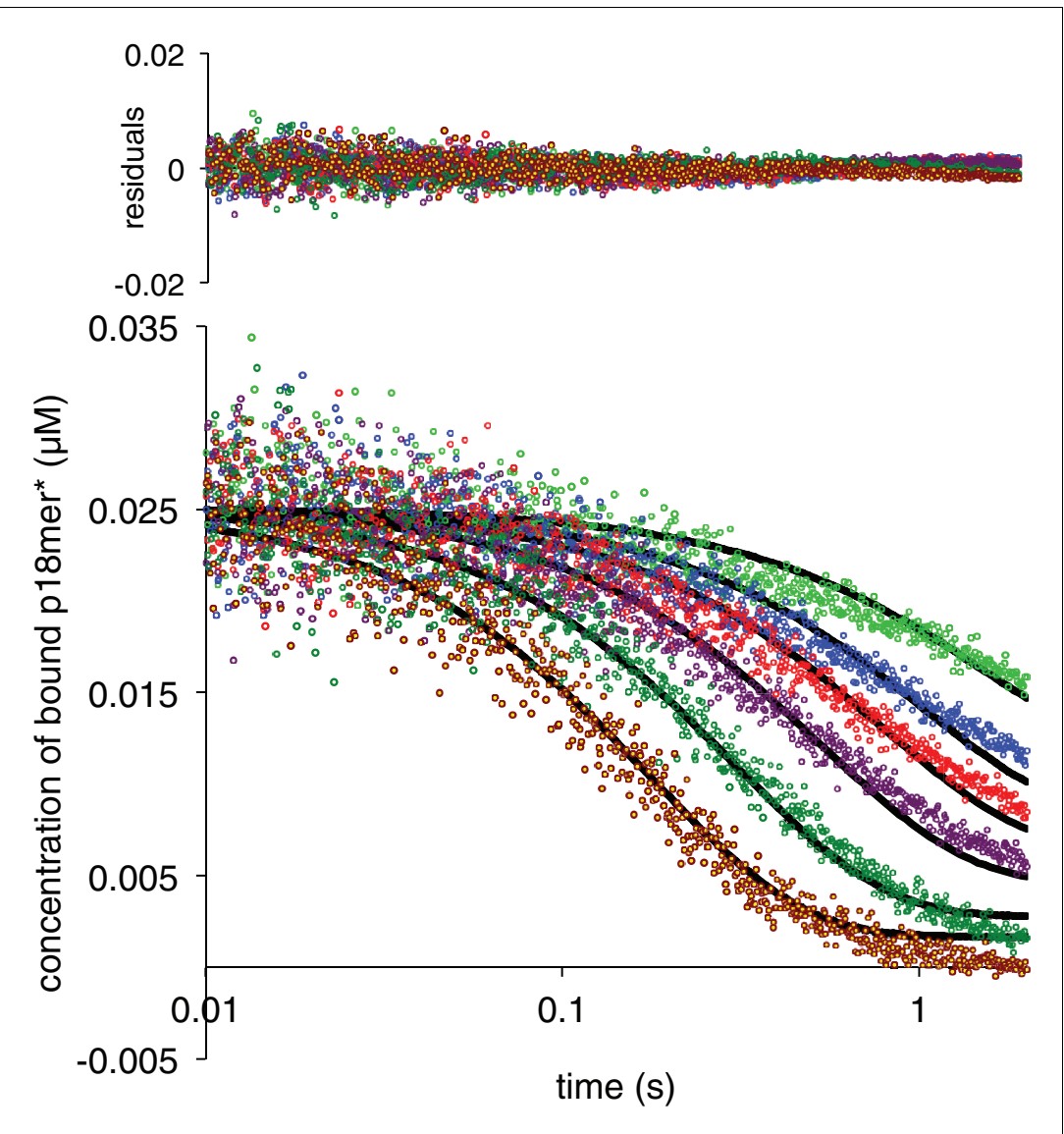

**Figure 4.** PARP1 dissociation from DNA as monitored by fluorescence anisotropy. Global fitting of six representative concentrations of competitor p18mer DNA using the mechanism in *Scheme 3*: 76 nM (light green), 149 nM (blue), 225 nM (red), 398 nM (violet), 1 μM (dark green), and 4 μM (brown). Residuals for the seven concentrations are shown overlaid in the corresponding colors above.

DOI: https://doi.org/10.7554/eLife.37818.027

The following source data and figure supplements are available for figure 4:

**Source data 1.** PARP1 dissociation from DNA as monitored by fluorescence anisotropy.

DOI: https://doi.org/10.7554/eLife.37818.038

**Figure supplement 1.** Thermodynamic parameters for DNA binding to PARP1.

DOI: https://doi.org/10.7554/eLife.37818.028

**Figure supplement 1—source data 1.** Thermodynamic parameters for DNA binding to PARP1.

DOI: https://doi.org/10.7554/eLife.37818.029

**Figure supplement 2.** PARP1 dissociation from DNA as monitored by fluorescence anisotropy.

DOI: https://doi.org/10.7554/eLife.37818.030

**Figure supplement 2—source data 1.** PARP1 dissociation from DNA as monitored by fluorescence anisotropy.

DOI: https://doi.org/10.7554/eLife.37818.031

**Figure supplement 3.** Dissociation of ΔZn1 from p18mer*.

DOI: https://doi.org/10.7554/eLife.37818.032

*Figure 4 continued on next page*

*Figure 4 continued*

**Figure supplement 3—source data 1.** Dissociation of ΔZn1 from p18mer*.
DOI: https://doi.org/10.7554/eLife.37818.033

**Figure supplement 4.** Dissociation of ΔZn2 from p18mer*.
DOI: https://doi.org/10.7554/eLife.37818.034

**Figure supplement 4—source data 1.** Dissociation of ΔZn2 from p18mer*.
DOI: https://doi.org/10.7554/eLife.37818.035

**Figure supplement 5.** Dissociation of ΔZn3 from p18mer*.
DOI: https://doi.org/10.7554/eLife.37818.036

**Figure supplement 5—source data 1.** Dissociation of ΔZn3 from p18mer*.
DOI: https://doi.org/10.7554/eLife.37818.037

Intact plasmid (4.5 kb;~90% supercoiled and 10% nicked) at 5 nM is a surprisingly effective competitor of a pre-formed 25 nM complex, yielding a $k_{obs}$ comparable to what is observed with 1 μM p18mer (*Figure 6A*). For comparison, 5 nM of p18mer yields no observable release of p18mer (*Figure 6A*). These results demonstrate that undamaged DNA is an effective trigger for the release of pre-bound p18mer*. To validate these results further, we prepared increasing numbers of free ends (models for DSBs) by restriction digests of the plasmid with different enzymes (*Figure 6—figure supplement 1*). If DNA ends are the actual triggers for release of pre-bound p18mer*, we would expect the purposeful increase in the numbers of ends (using the same amount of total plasmid) to yield increasing rates of release (see insert to *Figure 3*). Instead, we observe an unchanged $k_{obs}$, regardless of whether the concentration of ends is 10 nM (cut once), 20 nM (cut twice), 30 nM (cut thrice), or 240 nM (cut 24 times) (*Figure 6A*).

In the hallmark experiment of inter-strand transfer, we demonstrate the concentration-dependence of intact competitor plasmid on the apparent rate of release of p18mer*, wherein the data were analyzed using global fitting in Kintek Explorer. As seen for p18mer, increasing competitor DNA concentrations of plasmid DNA yielded increasing $k_{obs}$, and the data were best described by global fitting of the kinetic model of *Scheme 3* with the formation of a ternary complex (*Figure 6B*). In fact, the $k_{-2}$, $k_3$, and $k_{-3}$ are all similar to the values seen previously using p18mer (*Table 1*). In order to have 5 nM plasmid release all the 25 nM pre-bound p18mer*, we can assume there are minimally five binding sites per plasmid. At the other extreme, estimating that one can place one PARP1 every 10 bp, there are maximally ~450 binding sites. Thus, one can place limits on the true value for $k_2$ of 0.04–3.7 $nM^{-1}s^{-1}$. Interestingly, the lower limit of this rate of association is similar to that measured for p18mer (*Table 1*), suggesting that undamaged DNA is a very effective competitor of damaged DNA. We conclude that intact DNA can also engage the monkey-bar mechanism to facilitate the movement of PARP1 around the nucleus.

## The W589A mutant shows reduced accumulation at sites of DNA damage in cells

In order to test the physiological relevance of the mechanism of DNA-dependent release of DNA from PARP1 revealed in our in vitro experiments with p18mer and intact plasmid, we compared the rate and magnitude of accumulation of wild-type PARP1 with the W589A mutant at sites of DNA damage in cells. Mouse embryo fibroblasts were transiently transfected with GFP-tagged PARP1 (wild-type or W589A), and DNA damage was induced by laser microirradiation at a designated region of interest (ROI) within the nucleus. Accumulation of PARP1 in the ROI was monitored by confocal microscopy for 1–5 min and the diffusion coefficient ($D_{eff}$) and magnitude of PARP1 accumulation (F) were derived as recently described (*Mahadevan and Rudolph, 2018*). As seen in *Table 2*, the W589A mutant accumulated to a lower level and with a significantly slower diffusion coefficient than wild-type PARP1.

## High affinity inhibitors of PARP1 do not alter the rate or mechanism of DNA dissociation

Given the uncertain experimental basis for PARP1 trapping on DNA in cells treated with clinically relevant inhibitors of PARP1 (*Shen et al., 2015*), we used the rigorous in vitro assay described above

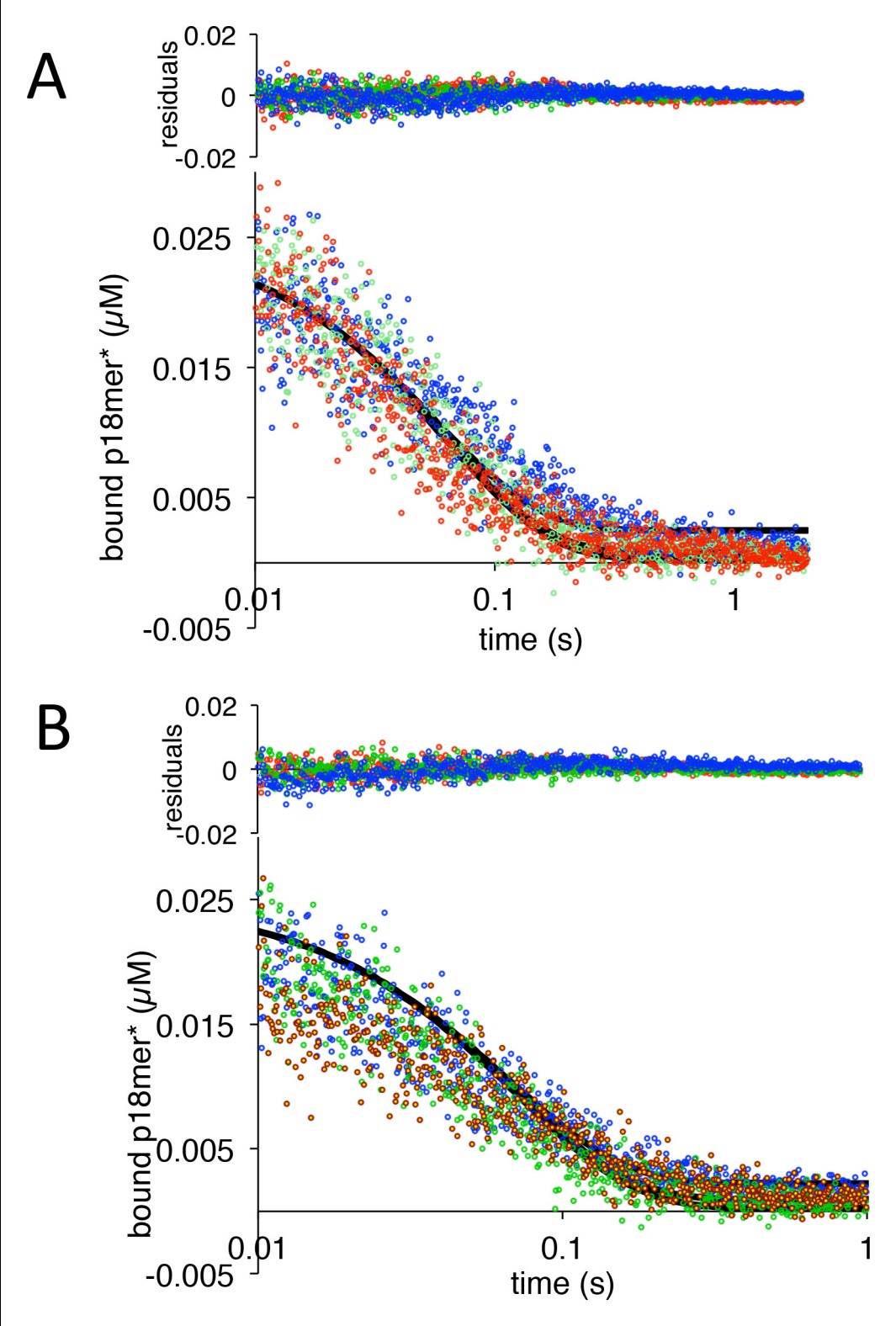

**Figure 5.** PARP1 dissociation from DNA as monitored by fluorescence anisotropy. Global fitting of three representative concentrations of competitor p18mer DNA using the mechanism in *Scheme 2*: 0.4 μM in blue, 1.3 μM in green, and 4 μM in red for (**A**) ΔWGR and (**B**) W589A. Residuals for the three concentrations are shown overlaid in the corresponding colors above.

DOI: https://doi.org/10.7554/eLife.37818.039

*Figure 5 continued on next page*

*Figure 5 continued*

The following source data is available for figure 5:

**Source data 1.** PARP1 dissociation from DNA as monitored by fluorescence anisotropy for ΔWGR from panel A.
DOI: https://doi.org/10.7554/eLife.37818.040
**Source data 2.** PARP1 dissociation from DNA as monitored by fluorescence anisotropy for W589A from panel B.
DOI: https://doi.org/10.7554/eLife.37818.041

to investigate whether these inhibitors lead to a change in the rate or mechanism of DNA dissociation. We monitored the dissociation of p18mer* from PARP1 by fluorescence anisotropy in the presence of four different tight-binding inhibitors of PARP1, using 1 µM competitor p18mer, conditions which lead to the inhibition of auto-PARylation (*Figure 7—figure supplement 1*). The observed dissociation curves were fit with a first-order exponential and the calculated rates were essentially identical to the DMSO control for olaparib, veliparib, niraparib, and talazoparib (*Figure 7A*). To ensure that in the presence of inhibitor, DNA dissociation was still dependent on binding of competitor DNA (*Scheme 3*), we investigated the DNA concentration dependence of p18mer* dissociation in the presence of talazoparib, the most potent PARP1-trapping compound (*Shen et al., 2015*). The dissociation of p18mer* in the presence of talazoparib (50 nM) was measured at varying concentrations of competitor p18mer (1–4 µM) and a concentration-dependent increase in $k_{obs}$ was observed just as for the control without inhibitor (*Figure 7B*). We conclude that these inhibitors do not change the rate or mechanism of DNA dissociation from PARP1.

## Discussion

Our results regarding the mechanisms of association and dissociation of PARP1 to and from DNA have important implications for our understanding of how PARP1 can move around the nucleus to scan for DNA damage. In vitro, PARP1 binds to DNA at or above the commonly accepted diffusion-limited rate (*Record et al., 1991*) of $1–2 \text{ nM}^{-1}\text{s}^{-1}$, consistent with its extremely rapid accumulation at sites of DNA damage in vivo (*Mortusewicz et al., 2007*; *Aleksandrov et al., 2018*). In fact, compared to other model systems using DNA oligomers, PARP1 – DNA association is among the fastest previously reported. For comparison, the rates of association of eukaryotic mismatch repair complex Msh2-Msh6 (*Biro et al., 2010*), RNA polymerase (*Johnson and Chester, 1998*), 8-oxoguanine-DNA-glycosylase (*Lukina et al., 2017*), and papillomavirus E2 protein (*Ferreiro and de Prat-Gay, 2003*)

**Table 2.** Fluorescence accumulation after DNA damage.
The differences in values of $D_{eff}$ and F (mean and standard error of the mean) were determined to be statistically significant using an unpaired t-test with p=0.0094 and p=0.016, respectively. The raw data for all the nuclei are provided as source data.

|  | Wild-type | W589A |
|---|---|---|
| $D_{eff}$ (µm$^2$/s) | 3.7 ± 0.6 | 2.1 ± 0.2 |
| F | 44 ± 3 | 33 ± 3 |
| n | 28 | 30 |

DOI: https://doi.org/10.7554/eLife.37818.046

The following source data is available for Table 2:
Source data 1. Fluorescence accumulation after DNA damage; values from the table.
DOI: https://doi.org/10.7554/eLife.37818.047

Source data 2. Normalized values for amount of fluorescence detected after DNA damage for each of 28 nuclei of WT PARP1.
DOI: https://doi.org/10.7554/eLife.37818.048

Source data 3. Normalized values for amount of fluorescence detected after DNA damage for each of 30 nuclei of W589A mutant.
DOI: https://doi.org/10.7554/eLife.37818.049

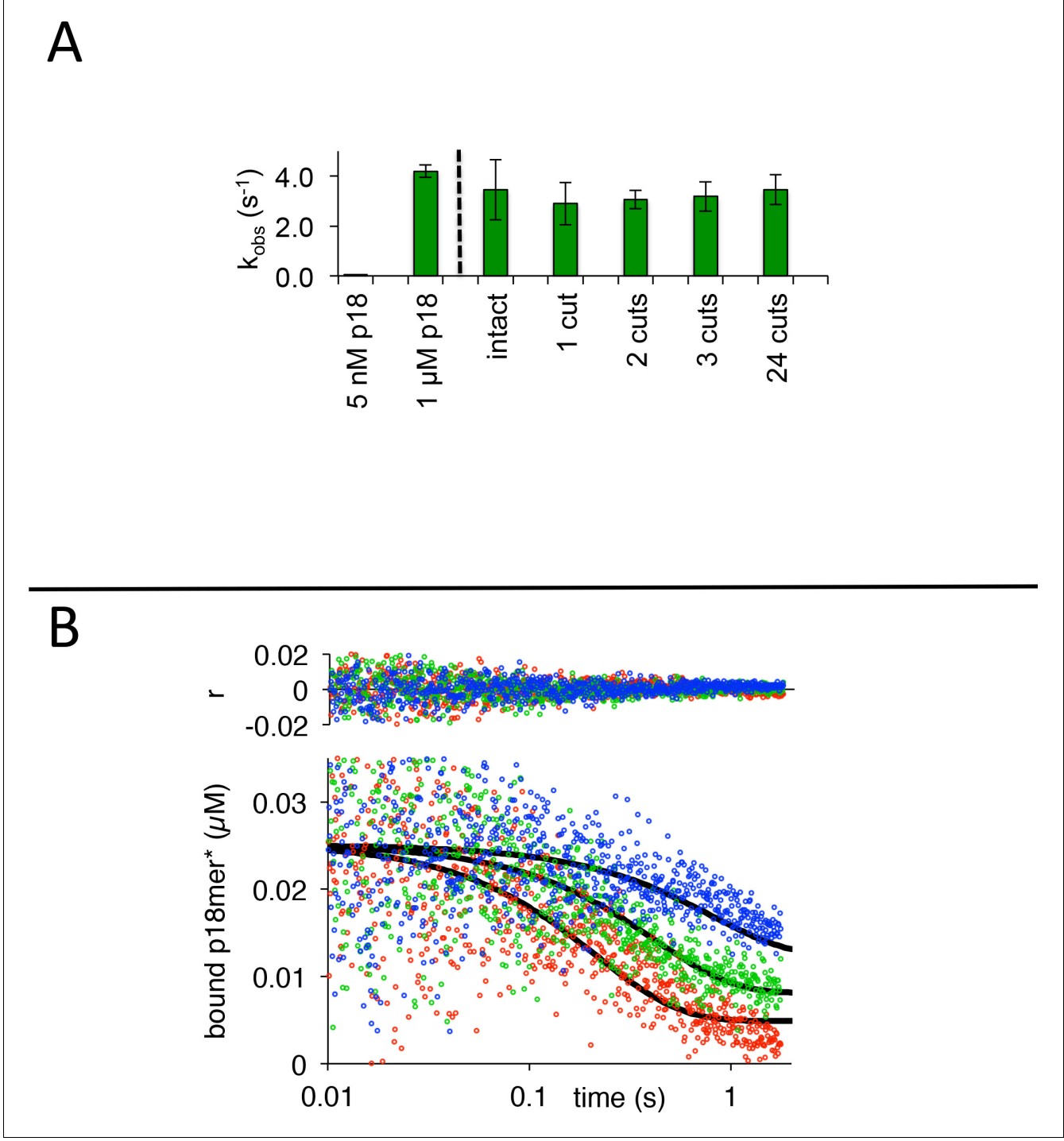

**Figure 6.** PARP1 dissociation from DNA as monitored by fluorescence anisotropy. (**A**) The observed rate of dissociation triggered by 5 nM of intact or variably cut plasmid is compared to 5 nM and 1 μM p18mer. (**B**) Global fitting of three representative concentrations of competitor intact plasmid DNA using the mechanism in *Scheme 3*: 0.7 nM in blue, 2.0 nM in green, and 5.8 μM in red. Residuals for the three concentrations are shown overlaid in the corresponding colors above.

DOI: https://doi.org/10.7554/eLife.37818.042

The following source data and figure supplement are available for figure 6:

**Source data 1.** The observed rate of dissociation triggered by 5 nM of intact or variably cut plasmid is compared to 5 nM and 1 μM p18mer.
DOI: https://doi.org/10.7554/eLife.37818.044

**Source data 2.** Global fitting of three representative concentrations of competitor intact plasmid DNA using the mechanism in *Scheme 3*.

*Figure 6 continued on next page*

*Figure 6 continued*

DOI: https://doi.org/10.7554/eLife.37818.045

**Figure supplement 1.** DNA 1% agarose gel demonstrating the various forms of plasmid used to demonstrate that undamaged DNA can effectively promote the monkey-bar mechanism for PARP1 to move from one segment of DNA to another.

DOI: https://doi.org/10.7554/eLife.37818.043

with double-stranded DNA are 0.002, 0.004, 0.13, and 0.6–1.4 $nM^{-1}s^{-1}$, respectively, effectively spanning three orders of magnitude. The fast association of PARP1 with DNA means that when dissociation does occur, re-association is most likely to the same site on the same DNA, as association is faster than diffusion carrying PARP1 away from its original binding site. This observation suggests that PARP1, like other DNA-binding proteins such as transcription factors (*Mirny et al., 2009*), must have a mechanism for moving around the genome that does not rely on simple dissociation and re-association. Although protein sliding along DNA in one-dimension has previously been invoked as a potential mechanism for accelerating the search for specific binding sites (*Berg et al., 1981*), more recent publications point out potential difficulties with such a long-distance sliding model (*Halford, 2009*), which is even more difficult to envision given the organization of DNA into nucleosomes in the eukaryotic genome.

Instead, we have found that PARP1 dissociation from DNA is triggered by binding of an additional DNA oligomer or undamaged plasmid prior to dissociation from the first DNA oligomer. We envision a 'monkey bar' model (*Vuzman et al., 2010a*; *Vuzman et al., 2010b*), wherein PARP1 moves from one DNA molecule to another DNA molecule, much like a child swings from bar to bar, transferring one hand at a time. This mechanism allows PARP1 to effectively scan the genome, moving to new and different sections of DNA. In the absence of competing DNA, PARP1 would remain effectively stuck at or near one site given its fast rate of association. Interestingly, undamaged DNA appears to be very effective at promoting the monkey-bar mechanism (*Table 1*), raising the question of how PARP1 remains stationary at sites of DNA damage. We have found that the WGR domain provides the other weaker 'hand' to facilitate the movement from one DNA strand to the next. Based on the structure of PARP1 bound to a DSB (*Langelier et al., 2012*), the WGR domain would need to first dissociate from the bound DNA prior to association with a second different molecule of DNA. Such a model is consistent with an NMR study that provides evidence for the stepwise assembly of PARP1 on a site of DNA damage where the Zn1, Zn2, and Zn3 domains engage a DNA molecule prior to final engagement of the WGR and catalytic domains (*Eustermann et al., 2015*). Thus, one can readily imagine a partial reversal of this process wherein the WGR domain releases the original DNA to bind the incoming DNA, followed by release of the original DNA and subsequent rearrangement of the domain around the newly bound DNA (*Figure 8*).

A recent study of PARP1 using single-molecule DNA tightrope assays provides strong evidence for such a monkey bar mechanism (*Liu et al., 2017*). It was shown that micro-dissociation of one of PARP1's multiple DNA-binding domains from DNA allows it to bind to a free 37 bp fragment, thus preventing rebinding of the domain to the tightrope and accelerating overall macro-dissociation. Our mechanism for DNA-dependent DNA dissociation also provides a compelling explanation for the wide diversity and significantly weaker dissociation constants previously reported for PARP1 with DNA (*Langelier et al., 2018*; *Clark et al., 2012*; *Langelier et al., 2010*): the measured apparent $K_D$ depends strongly on the experimental conditions (i.e. [DNA]) under which the experiment is performed, since higher concentrations of DNA promote release of DNA.

Strong in vitro evidence for a monkey bar mechanism, also known as intersegment transfer, exists for various other proteins, all of them transcription factors. Kinetic experiments similar to ours have demonstrated strong dependence of DNA release on additional DNA for the *lac* repressor (*Ruusala and Crothers, 1992*), cAMP receptor protein (*Fried and Crothers, 1984*) and glucocorticoid receptor (*Lieberman and Nordeen, 1997*). NMR methods have demonstrated that both HoxD9 (*Iwahara et al., 2006*) and Oct1 (*Doucleff and Clore, 2008*) can bridge one DNA strand to another. A combination of methods including NMR, rate measurements, and computational modeling have elegantly demonstrated how EgrI uses its three Zn fingers to both bind and scan other DNA fragments as it moves between different recognition sites on different DNA molecules (*Zandarashvili et al., 2015*). Interestingly, mutagenesis of specific residues in EgrI was used to shift the equilibrium between binding and scanning modes.

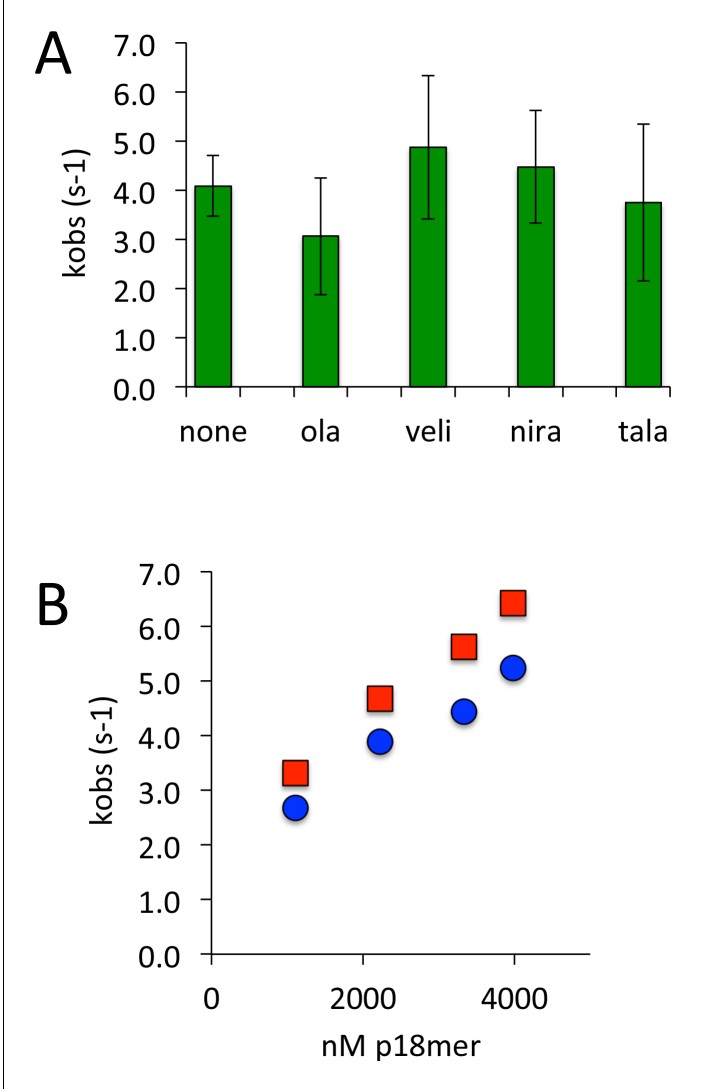

**Figure 7.** PARP1 dissociation from DNA as monitored by fluorescence anisotropy in the presence of various inhibitors of PARP1. (**A**) Apparent $k_{obs}$ at 1 μM competitor DNA using four different inhibitors (50 nM) (ola = olaparib, veli = veliparib, nira = niraparib, tala = talazoparib). (**B**) Apparent $k_{obs}$ at 1–4000 nM competitor DNA for PARP1 alone (red) and PARP1 the presence of 50 nM talazoparib (blue).

DOI: https://doi.org/10.7554/eLife.37818.050

The following source data and figure supplement are available for figure 7:

**Source data 1.** PARP1 dissociation from DNA as monitored by fluorescence anisotropy in the presence of various inhibitors of PARP1.

DOI: https://doi.org/10.7554/eLife.37818.052

**Figure supplement 1.** Smear assay of PARP1 demonstrating the effectiveness of four different inhibitors in blocking the activity of PARP1.

DOI: https://doi.org/10.7554/eLife.37818.051

One major caveat to any biochemical investigation of the mechanism of DNA – protein interactions is the artificial nature of a DNA oligomer compared to intact chromatin. This limitation affects studies of PARP1 interactions with DNA in particular because it is quite difficult to prepare completely intact DNA without ends or nicks, preferably wrapped around nucleosomes. Thus, it was important to test the significance of the monkey bar mechanism in a more physiologically relevant model. We have demonstrated the validity of interstrand transfer in vivo by demonstrating that the point mutant W589A, which disrupts DNA-dependent release of DNA, accumulates slower and to a

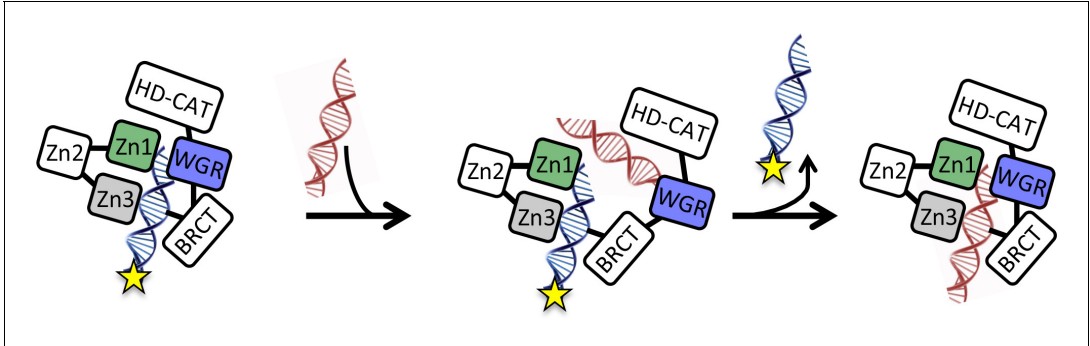

**Figure 8.** Model for the monkey bar mechanism for PARP1 depicting the proposed role of the WGR domain in capturing the second DNA strand prior to release of the originally bound DNA strand and subsequent re-arrangement around the second DNA.
DOI: https://doi.org/10.7554/eLife.37818.053

lesser amount than wild-type PARP1 at sites of laser microirradiation (*Table 2*). The slower accumulation of W589A in cells is a particularly powerful demonstration of the importance of the monkey bar mechanism for PARP1 in finding sites of DNA damage for two reasons. First, the rate of dissociation for W589A from DNA is greater than for wild-type ($20\,s^{-1}$ vs. $< 10\,s^{-1}$). Second, the apparent $K_D$ of W589A for DNA is weaker than that of wild-type PARP1 (5 nM vs. < 3 nM, respectively). Simplistically, these two observations might suggest that W589A should arrive at sites of DNA damage more rapidly than wild-type PARP1 since its interaction with DNA is not as tight or as long-lived (i.e. it spends less time occupying irrelevant sites); yet we observe the opposite. The monkey bar mechanism provides the explanation for these results: high concentrations of intranuclear DNA allow PARP1 to explore the nucleus rapidly. A dysfunctional monkey (W589A) surrounded by a lot of DNA does not move as rapidly. Thus, we have quantitatively demonstrated the importance of intersegment transfer in the accumulation of a DNA-binding protein at its target destination in vivo.

Finally, our results also provide further insight into the much-discussed topic of PARP1 'trapping', wherein cells treated with inhibitors of PARP1 exhibit a shift of PARP1 from the soluble fraction to a chromatin-associated insoluble fraction (*Murai et al., 2012*). Many cell-based studies have since confirmed the phenomenon of PARP-trapping (*Pommier et al., 2016*). Our data showing no effect of four different tight-binding inhibitors of PARP1 on the release of DNA agree with a previous thorough biochemical analysis that also could not find any effects of inhibitors on DNA binding constants or rates of dissociation (*Hopkins et al., 2015*). Thus, the mechanistic basis for PARP-trapping is more complex than can be reconstituted in vitro. Our results are in agreement with recent findings that PARP inhibitors lead to defective fork recovery and/or homologous recombination-mediated repair, and thus an increase in DNA damage where PARP1 is bound to DNA, and lacking activity due to inhibitor binding (*Maya-Mendoza et al., 2018*).

## Materials and methods

### Key resources table

| Reagent type (species) or resource | Designation | Source or reference | Identifiers | Additional information |
|---|---|---|---|---|
| Cell line (*Mus musculus*) | Mouse embryo fibroblasts (MEF) | Dantzer Lab, University of Strasbourg, France | | cells routinely tested negative for Mycoplasma contamination by Cell Culture Facilities |
| Transfected construct (*Homo sapiens*) | pEGFP-C3-PARP1 | Dantzer Lab, University of Strasbourg, France | | |
| Peptide, recombinant protein | PARP1 | Pascal Lab, University of Montreal | | |

*Continued on next page*

*Continued*

| Reagent type (species) or resource | Designation | Source or reference | Identifiers | Additional information |
|---|---|---|---|---|
| Chemical compound, drug | NAD+ | Sigma | N1511 | 34 mM stock solution; pH'd to ~ 7 |
| Chemical compound, drug | olaparib | Selleck | S1060 | 10 mM stock solution in DMSO |
| Chemical compound, drug | veliparib | Selleck | S1004 | 10 mM stock solution in DMSO |
| Chemical compound, drug | niraparib | Selleck | S2741 | 10 mM stock solution in DMSO |
| Chemical compound, drug | talazoparib | Selleck | S7048 | 10 mM stock solution in DMSO |
| Chemical compound, drug | JetPei | Polyplus Transfection | 101–40N | |
| Chemical compound, drug | Hoechst 33342 | InVitrogen | H1399 | |
| Software, algorithm | MatLab code | doi.org/10.1101/373043 | DNA Repair Analysis Toolbox.mltbx | for conversion of microscope data to text files |
| Software, algorithm | MatLab code | doi.org/10.1101/373043 | Bioformats Toolbox (v1.0.4b).mltbx | need as an accessory for the DNA Repair Analysis Toolbox |
| Software, algorithm | Mathematica code | doi.org/10.1101/373043 | | for analysis of microscope text files by free diffusion model |
| Software, algorithm | Kintek Explorer | http://kintek-corp.com/ | | for analysis of kinetic data |

## Materials

NAD$^+$ was obtained from Sigma. Olaparib, veliparib, niraparib, and talazoparib were obtained from Selleck. DNA oligonucleotides and their complementary strands were obtained from IDT: p18mer: 5'-phosphate-GGGTTGCGGCCGCTTGGG-3'. Labeled oligonucleotides with a 5'-fluorescein dye (*) were also obtained from IDT. Double-stranded fragments were prepared by annealing at 100 μM DNA in 10 mM Tris-HCl (pH 7.5), 100 mM NaCl, and 0.1 mM EDTA. The DNA was heated to 95°C for 5 min and then slowly cooled at 0.1 °C/second to 4°C. Annealing was confirmed by 10% (wt/vol) native TBE-PAGE at 200 V for 30 min. Intact supercoiled plasmid (pUC19-601-147-12copy) is a pUC derivative that was prepared as described (*Dyer et al., 2004*). Restriction enzymes were obtained from New England Biolabs.

## Cloning of deletion constructs ΔZn1, ΔZn2, ΔZn3, and ΔWGR of PARP1

The pET28a vector encoding cDNA of full-length human PARP1 was used to design constructs lacking various domains of PARP1 following the method outlined in Hansson et al (*Hansson et al., 2008*). Briefly, primer one was designed as a reverse complement of the sequence that corresponds to 20–25 bases upstream of the DNA sequence to be deleted, followed by 20–25 bases corresponding to the downstream sequence. Primer two corresponds to the complementary strand. These primers were used in a PCR reaction to loop out the DNA encoding individual domains of PARP1: ΔZn1 (M1-K97), ΔZn2 (G96-linker-K207), Δn3 (G215-linker-A367), and ΔWGR (N517-linker-L655). After PCR, DpnI digestion was used to degrade the template plasmid and was then transformed to generate clones. Next, a linker DNA sequence encoding amino acids LLA(GS)$_4$GAAL was inserted in place of the deleted domain using partially overlapping primers comprising the entire sequence of the insert followed by 20–25 bases of the downstream sequence. Thereafter, another step of insertion of linker DNA sequence encoding amino acids ALA (GS)$_5$GLAL upstream of the previous insert was performed in a similar manner. The plasmids used to express various domain deletion PARP1 mutants eventually all contained the 30 amino acid linker ALA (GS)$_5$GLALLLA(GS)$_4$GAAL in place of the deleted PARP1 domain. The W589A mutant of PARP1 was generated using QuikChange

Mutagenesis (Agilent) following the manufacturer's instructions. All constructs were verified by DNA sequencing of the entire PARP1 gene.

## Expression and purification of PARP1

Wild-type PARP1, all deletion constructs, and the W589A mutant of PARP1 were expressed and purified from *E. coli* as previously described (*Clark et al., 2012*; *Langelier et al., 2011b*) with the minor modification that PARP1 was eluted from the nickel-NTA column using a gradient from 20 to 400 mM imidazole.

## Activity and stability measurements of PARP1

PARylation activity was evaluated by incubating 0.5 µM PARP1 with 1 µM p18mer and 500 µM $NAD^+$ in 50 mM Tris- HCl (pH 7.5), 50 mM NaCl, and 1 mM $MgCl_2$, for 5 min. Reactions were quenched in Laemmli buffer, boiled for 5 min, and then resolved on SDS-PAGE (4–20%). PARP1 stability was evaluated using the Protein Thermal Shift Dye Kit from Applied Biosystems and a BioRad C1000 ThermalCycler with a CFX96 RealTime module.

## Stopped-flow fluorescence anisotropy

A SX20 Stopped-Flow Spectrometer (Applied Photophysics) was used for measuring fluorescence anisotropy using an excitation wavelength of 485 nm and cut-off filters in the parallel and perpendicular detectors at 515 nm. Association reactions were measured by mixing equal volumes of p18mer* (60 nM) with three to eight different concentrations of PARP1 (60–250 nM) and monitoring the anisotropy at 20°C for 25 ms. All indicated concentrations are after mixing. Although PARP1 can bind to both ends of p18mer* (and p18mer) simultaneously (*Langelier et al., 2012*), we treat each DNA oligomer as one equivalent (not two) because fluorescence anisotropy detects only the first binding event. Control reactions used for determining background signal lacked PARP1. For measuring dissociation, a pre-formed complex of PARP1 (37 nM) and p18mer* (25 nM) was mixed with 5–15 different concentrations of p18mer (100 nM – 8 µM) and anisotropy was monitored at 20°C for 1–5 s. Control reactions for determining background signal lacked p18mer. All reactions were performed in 50 mM Tris-HCl (pH 7.5), 50 mM NaCl, 1 mM $MgCl_2$, 0.1 mM EDTA, and 0.01% IGEPAL. For all stopped-flow reactions, data were collected in log mode, and 10–12 shots were averaged for each different concentration of reagents. All experiments consisting of series of different concentrations of PARP1 (for association) or of p18mer (for dissociation) were performed on at least three different days with a least two different preparations of protein. Plasmids for dissociation experiments were either untreated or digested at 0.25 mg/mL using 50–1000 U of the appropriate restriction enzyme at 37°C overnight. SacI, DrdI, EarI, and EcoRV were used to generate 1, 2, 3, and 24 cuts, respectively. (Digestion with EcoRV yields the parent plasmid and 12 identical inserts of 147 bp DNA). Dissociation experiments in the presence of inhibitors (50 nM) of PARP1 were compared to DMSO controls (<2 %v/v).

## Data fitting

Data were initially analyzed for fitting to single exponential kinetics using the software from Applied Photophysics (ProDataTSV). Global analysis incorporating multiple different concentrations of protein or competing DNA were performed using KinTek Explorer (KinTek Corporation). For association kinetics, control reactions in the absence of protein were used to determine the baseline, and the maximum anisotropy signal (identical at all protein concentrations) was used to convert anisotropy units to concentration values. For dissociation kinetics, control reactions in the absence of DNA were used to determine the baseline, and the maximum anisotropy signal at high concentrations of p18mer (1–4 µM) were used to convert anisotropy units to concentration values. For dissociation kinetics using plasmid, global fitting was performed by assuming 5–450 binding sites/4.5 kb plasmid, which as consequence implies that the value of $k_2$ is subject to this assumption (*Scheme 3*). However, values for $k_{-2}$, $k_3$, and $k_{-3}$ are independent of this assumption. For clarity, only a subset of the concentrations used experimentally are shown in the figures.

## Quantitation of fluorescence accumulation of GFP-PARP1 after laser microirradiation

Mammalian expression plasmid (pEGFP-C3, 250 ng/20,000 cells) encoding full-length GFP-tagged human PARP1 was transfected using jetPEI (Polyplus Transfection) into wild-type mouse embryo fibroblasts (MEFs) cultured in DMEM supplemented with 50 µg/ml of gentamicin and 10% FBS, as previously described (Mahadevan et al, in preparation). Cells were sensitized with Hoechst 33342 (Invitrogen) (10 µg/ml) for 10 min prior to induction of DNA damage using a 405 nm diode laser (100% power for 1 s). Cells were imaged for 1–5 min using excitation at 488 nm within a heated environmental chamber set to 37°C and 5% $CO_2$ (Nikon A1R confocal laser scanning; frame size of 512 × 512). Analysis of the fluorescent images was carried out using custom codes in MATLAB and Mathematica to allow derivation of the diffusion coefficient ($D_{eff}$) and mobile fraction of PARP1 (F) (*Mahadevan and Rudolph, 2018*). These codes are provided as DNA Repair Analysis Toolbox.mtlbx (*Source code 1*), Bioformats Image Toolbox (v1.0.4b).mltbx (*Source code 2*), and Q-FADD 0.13.nb (*Source code 3*).

## Acknowledgements

This work was supported by NIH-NCI R01 CA218255, the University of Colorado Cancer Center Pilot Funding Grant ST63501792, and the Howard Hughes Medical Institute.

## Additional information

### Funding

| Funder | Grant reference number | Author |
| --- | --- | --- |
| National Cancer Institute | R01 CA218255 | Karolin Luger |
| Howard Hughes Medical Institute | | Karolin Luger |
| University of Colorado Cancer Center | Pilot Funding Grant ST63501792 | Karolin Luger |

The funders had no role in study design, data collection and interpretation, or the decision to submit the work for publication.

### Author contributions

Johannes Rudolph, Conceptualization, Data curation, Formal analysis, Validation, Methodology, Writing—original draft, Writing—review and editing; Jyothi Mahadevan, Formal analysis, Validation, Investigation, Methodology; Pamela Dyer, Investigation; Karolin Luger, Conceptualization, Funding acquisition, Project administration, Writing—review and editing

### Author ORCIDs

Johannes Rudolph http://orcid.org/0000-0003-0230-3323
Pamela Dyer http://orcid.org/0000-0002-0142-5073
Karolin Luger http://orcid.org/0000-0001-5136-5331

### Decision letter and Author response

Decision letter https://doi.org/10.7554/eLife.37818.059
Author response https://doi.org/10.7554/eLife.37818.060

## Additional files

### Supplementary files

- Source code 1. DNA repair analysis toolbox
DOI: https://doi.org/10.7554/eLife.37818.054
- Source code 2. Bioformats image toolbox

DOI: https://doi.org/10.7554/eLife.37818.055
• Source code 3. Q-FADD 0.13.nb
DOI: https://doi.org/10.7554/eLife.37818.056
• Transparent reporting form
DOI: https://doi.org/10.7554/eLife.37818.057

## Data availability

We have provided Excel files for all the figures and tables.

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
