## [Decision Letter]

Thank you for submitting your article "Poly(ADP-ribose) polymerase 1 Searches DNA via a 'Monkey Bar' Mechanism" for consideration by *eLife*. Your article has been favorably reviewed by three peer reviewers, including Wolf-Dietrich Heyer as the Reviewing Editor and Reviewer #1, and the evaluation has been overseen by John Kuriyan as the Senior Editor. The following individual involved in review of your submission has agreed to reveal their identity: Peter von Hippel (Reviewer #2).

The reviewers have discussed the reviews with one another and the Reviewing Editor has drafted this decision to help you prepare a revised submission.

This is a well-written manuscript that analyzes an important problem of how PARP1 finds DNA damage in the genome and how PARP inhibition affects PARP1 DNA binding. The authors show that dissociation of PARP1 from DNA requires another DNA molecule in a process that requires the WGR domain of PARP1. The work represents a very nice piece of physical biochemical analysis that provides significant support for the overall transfer scheme outlined in Figure 7. The requirement for the WGR domain was convincingly shown by using mutant variants (deletion, point mutation) and the biological significance of the biochemical observations was validated using wild type and mutant PARP1-GFP fusion constructs in fibroblast cells. Finally, the authors use their biochemical DNA dissociation assay to study the mechanism of action of PARP1 inhibitors, which have been postulated to trap PARP1 in its DNA-bound form. The authors do not detect any effect of the PARP1 inhibitor talazoparib on DNA dissociation in their experimental system.

Major concerns that are critical to address in a revision:

1) The authors use an 18mer oligonucleotide substrate to study the association of PARP1 to undamaged DNA and the transfer to another stretch of undamaged DNA. The concern is that this substrate does not mimic undamaged DNA but rather a DSB. Hence, the authors do not study, as implied in the title and clearly stated in the manuscript, PARP1 searching for damaged DNA but PARP1 exchange from damaged DNA to another piece of damaged DNA. The key experiment that would need to be added is binding to a small circular DNA and competition by another undamaged circular DNA to validate the key observation of competitor-induced dissociation for a search modus on undamaged DNA.

These additional experiments appear feasible and represent the preferred option for publication with the current claims in the title and descriptions regarding the search modus on undamaged DNA. Alternatively, in case such experiments cannot be conducted for technical reasons or not be completed in a reasonable time frame, we suggest that the title and the description in the text be modified to acknowledge the limitations of the model substrates for extrapolating to a genome-wide search for DNA damage.

It should be noted that the rapid dissociation of PARP1 in the presence of competitor DNA has previously been observed using a surface plasmon resonance (SPR) biosensor (PMID 19585541) and fluorescence anisotropy (PMID: 29487285). The second study is referenced, but the SPR study is not. The SPR study provides on and off rates that would be useful to compare to the present analysis. The current study goes on to identify the WGR domain as the principle element responsible for the release of PARP1, so there is a clear advance of the present work.

2) The data from the in vivo experiments measuring PARP1 accumulation at laser-induced DNA damage sites in Table 2 cannot be fully evaluated, as the description is very brief. The authors refer to a manuscript in preparation, which is not accessible. While the authors decided apparently to publish the method elsewhere, the present manuscript requires a fuller description of the method and a more detailed presentation of the data including examples of the fluorescent images and kymographs. As the in vivo data provide the biological significance of the biochemical data, this part is integral to the manuscript.

The slower diffusion and lower accumulation of the WGR mutant are difficult to interpret only in terms of the monkey bar mechanism. For example, the mutant is shown to be catalytically inactive, which could potentially influence its mobility through the nucleus. The retention of the mutant at damage sites might be lower simply due to its lower affinity for DNA damage. It should be noted that there is some evidence in the literature that PARP1 activity can influence the localization to sites of damage (PMID: 17982172). At least the text should address this idea.

In order to expand the description of the cell-based experiments the authors are welcome to add additional authors if that is needed to facilitate the expansion of this aspect.

Additional points:

3) The experiments are conducted at a rather low salt concentration (significantly less than physiological). Low salt would tend to strengthen non-specific binding interactions and perhaps "hide" specific binding interactions. Have experiments been conducted at higher salt concentrations, and if so please comment on any differences or complications observed? If the experiment still works it might even strengthen the key role of the WRG domain, and perhaps provide an opening for the clinical inhibitors to work.

4) Although the term "monkey bar" mechanism for some forms of inter-segment transfer in such 'facilitated diffusion' experiments has been used before (see Vuzman, Azia and Levy, 2010), it is not clear whether it is an improvement over the somewhat less graphic and more general 'inter-segment transfer' designation, especially in the title of the article. We ask the authors whether, and if so why, they think it is especially appropriate here.

5) In the fourth paragraph of the Discussion, where the authors discuss other studies in which intersegment transfer mechanisms have been shown to be important, they might consider citing also that might be the earliest such study that focused primarily on this mechanism using the lac repressor system by Ruusala and Crothers (1992).

6) Scheme 2, last complex should be P1-DNA (no star).

7) "whereas the Y-intercept equals the first-order rate constant of dissociation"

This section was difficult to follow for the non-specialist. The plot being referenced has an apparent Y-intercept of zero. I believe this is meant to indicate that the data points with PARP-1 protein included yield a line that extends toward a crossing point on the Y-axis, and in this case it is a very small number but above zero, since the dissociation is anticipated to be very small. However, it seems that the zero concentration point of PARP-1 is included in the line fit, which would seem to contribute to moving the line toward the origin of the plot. Should the zero concentration of PARP-1 be included?

8) "as previously reported (Clark et al., 2012), Zn1, Zn3, and WGR are essential for catalytic activity…" I was not able to find this data in the reference provided, perhaps a problem with the citations.

---

## [Author Response]

Major concerns that are critical to address in a revision:1) The authors use an 18mer oligonucleotide substrate to study the association of PARP1 to undamaged DNA and the transfer to another stretch of undamaged DNA. The concern is that this substrate does not mimic undamaged DNA but rather a DSB. Hence, the authors do not study, as implied in the title and clearly stated in the manuscript, PARP1 searching for damaged DNA but PARP1 exchange from damaged DNA to another piece of damaged DNA. The key experiment that would need to be added is binding to a small circular DNA and competition by another undamaged circular DNA to validate the key observation of competitor-induced dissociation for a search modus on undamaged DNA.These additional experiments appear feasible and represent the preferred option for publication with the current claims in the title and descriptions regarding the search modus on undamaged DNA. Alternatively, in case such experiments cannot be conducted for technical reasons or not be completed in a reasonable time frame, we suggest that the title and the description in the text be modified to acknowledge the limitations of the model substrates for extrapolating to a genome-wide search for DNA damage.

We appreciate the reviewers’ request to study PARP1 binding to intact DNA, and PARP1 then moving to other intact DNA. In fact, we have spent a considerable amount of time attempting to find a convincing model for intact DNA that still allows for measurement of PARP1 binding. However, we discovered that any attempt to label intact DNA or end-cap shorter pieces of DNA to hide the ends from PARP1-binding appears to mimic DNA damage. Since the key question the reviewers ask is whether undamaged DNA can act as an effective competitor, i.e. allow PARP1 to move around the genome via inter-strand transfer, we have designed and performed the experiments described in the subsection “Undamaged DNA also facilitates the monkey-bar mechanism”, with a new Figure 6 and Figure 6—figure supplement 1. Briefly, we show that (1) a large mostly supercoiled plasmid that has a high “intact middle” but vanishingly small “end/nick” concentration can act as a surprisingly effective competitor to promote release of p18mer*; (2) varying the concentration of supercoiled plasmid yields the same concentration-dependence seen for competition experiments with p18mer; and (3) increasing the number of ends in the same plasmid by introducing various number of restriction enzyme cuts does not lead to a change in the effectiveness of competition. That is, if the ends were promoting the competition, then the same amount of a more frequently cut plasmid should yield a more effective competitor, which is not the case. We conclude that intact DNA can also act to promote inter-strand transfer.

It should be noted that the rapid dissociation of PARP1 in the presence of competitor DNA has previously been observed using a surface plasmon resonance (SPR) biosensor (PMID 19585541) and fluorescence anisotropy (PMID: 29487285). The second study is referenced, but the SPR study is not. The SPR study provides on and off rates that would be useful to compare to the present analysis. The current study goes on to identify the WGR domain as the principle element responsible for the release of PARP1, so there is a clear advance of the present work.

SPR methods are often questionable for studying true diffusion-limited interactions because of the mass transport problem near the surface (see for example PMID 20217612). In fact, the measured rates of association in PMID 19585541 are 2 – 3 orders of magnitude slower than what we measure in solution. The rates of dissociation in the SPR study are difficult to compare as they do not use competitor DNA, but they are very slow (3 – 7) x 10^-3^ s^-1^. We have added a literature reference for completeness. No quantitation is presented in PMID 29487285, so comparisons are difficult.

2) The data from the in vivo experiments measuring PARP1 accumulation at laser-induced DNA damage sites in Table 2 cannot be fully evaluated, as the description is very brief. The authors refer to a manuscript in preparation, which is not accessible. While the authors decided apparently to publish the method elsewhere, the present manuscript requires a fuller description of the method and a more detailed presentation of the data including examples of the fluorescent images and kymographs. As the in vivo data provide the biological significance of the biochemical data, this part is integral to the manuscript.We provide a link to our detailed methods in the form of a BioRxiv article (Mahadevan et al., 2018) that we have submitted to Nature Methods. We have also increased the number of analyzed nuclei since our original submission with no changes in the statistical outcome.The slower diffusion and lower accumulation of the WGR mutant are difficult to interpret only in terms of the monkey bar mechanism. For example, the mutant is shown to be catalytically inactive, which could potentially influence its mobility through the nucleus. The retention of the mutant at damage sites might be lower simply due to its lower affinity for DNA damage. It should be noted that there is some evidence in the literature that PARP1 activity can influence the localization to sites of damage (PMID: 17982172). At least the text should address this idea.First, we note that while it may be correct to speculate that the lower retention at the sites of damage could be due to a lower affinity for DNA, it would be incorrect to ignore the *slower* arrival of the WGR mutant at sites of DNA damage, which is the result we most focus on regarding the mechanism of PARP1 movement inside the nucleus (see Discussion). The reviewers are correct in pointing out that the activity of PARP1 can potentially influence the localization of PARP1 in vivo. In fact, PMID17982172 actually shows that the activity-deficient mutant accumulates to a higher level than WT PARP1, in contrast to the lower level we observe for the W589A mutant. The actual rates of accumulation are not quantitated in PMID17982172 and difficult to interpret from the averaging of differently shaped nuclei (see Mahadevan et al., 2018).In order to expand the description of the cell-based experiments the authors are welcome to add additional authors if that is needed to facilitate the expansion of this aspect.

The availability of our detailed methods in the bioRxiv article (Mahadevan et al., 2018) fulfills the reviewers’ request.

Additional points:3) The experiments are conducted at a rather low salt concentration (significantly less than physiological). Low salt would tend to strengthen non-specific binding interactions and perhaps "hide" specific binding interactions. Have experiments been conducted at higher salt concentrations, and if so please comment on any differences or complications observed? If the experiment still works it might even strengthen the key role of the WRG domain, and perhaps provide an opening for the clinical inhibitors to work.

The total ion concentration used in our experiments is 102 mM. Although this is somewhat lower than what is considered physiological (150 mM), it is not hugely different, unlike many earlier experiments in the protein – DNA literature that were performed at < 10 mM total ion concentration. We have performed some experiments at higher salt concentrations, and find that on- and off-rates do not change significantly with the addition of 100 – 200 mM additional NaCl to our current buffer conditions (not shown). It is unclear what the reviewer means by “for the clinical inhibitors to work”: we find that the inhibitors bind to PARP1 and inhibit its activity under the buffer conditions used. We have added Figure 7—figure supplement 1 to show this inhibition. They do not, however, trap DNA on PARP1, as has also been shown by others (see Hopkins et al., 2015). We have also added a reference to a recent Nature paper (Maya-Mendoza et al., 2018, PMID 29950726) to the Discussion providing an explanation why PARP-trapping is seen in vivo, but not in vitro.

4) Although the term "monkey bar" mechanism for some forms of inter-segment transfer in such 'facilitated diffusion' experiments has been used before (see Vuzman, Azia and Levy 2010), it is not clear whether it is an improvement over the somewhat less graphic and more general 'inter-segment transfer' designation, especially in the title of the article. We ask the authors whether and if so why they think it is especially appropriate here.

The term monkey-bar gives the reader an immediate, very intuitive, understanding of the major results of this manuscript. Also, one of the difficulties encountered in the “Google Age” is that everything is so readily accessible by a simple web search, but the number of returned hits is so large that the desired results are actually rather difficult to find. Thus we believe it is important to use unique words that provide an immediate visual understanding. There is significant precedent in the field for using the term monkey bar, including links to helpful explanatory videos upon performing a Google search using the terms monkey bar, protein, and DNA.

5) In the fourth paragraph of the Discussion, where the authors discuss other studies in which intersegment transfer mechanisms have been shown to be important, they might consider citing also that might be the earliest such study that focused primarily on this mechanism using the lac repressor system by Ruusala and Crothers (1992).Added. Thanks for pointing out that additional reference.6) Scheme 2, last complex should be P1-DNA (no star).

Corrected. Thank you.

7) "whereas the Y-intercept equals the first-order rate constant of dissociation"This section was difficult to follow for the non-specialist. The plot being referenced has an apparent Y-intercept of zero. I believe this is meant to indicate that the data points with PARP-1 protein included yield a line that extends toward a crossing point on the Y-axis, and in this case it is a very small number but above zero, since the dissociation is anticipated to be very small. However, it seems that the zero concentration point of PARP-1 is included in the line fit, which would seem to contribute to moving the line toward the origin of the plot. Should the zero concentration of PARP-1 be included?

The reviewer makes a very good point. We have repaired the graph: inclusion of the 0,0 point was not intentional.

8) "as previously reported (Clark et al., 2012), Zn1, Zn3, and WGR are essential for catalytic activity…" I was not able to find this data in the reference provided, perhaps a problem with the citations.

Yes, this was a typo in the citations. Thank you for finding it. The correct reference is Langelier et al. (2012).